# Development of a non-infectious control for viral hemorrhagic fever PCR assays

**Matthew A. Knox** [1]*, **Collette Bromhead**[2], **David TS Hayman**[1]

1 Massey University, School of Veterinary Science, Palmerston North, Manawatu-Wanganui, New Zealand,
2 Massey University, School of Health Sciences, Wellington, New Zealand

* M.Knox@massey.ac.nz

## Abstract

Assay validation is an essential component of disease surveillance testing, but can be problematic in settings where access to positive control material is limited and a safety risk for handlers. Here we describe a single non-infectious synthetic control that can help develop and validate the PCR based detection of the viral causes of Crimean-Congo hemorrhagic fever, Ebola virus disease, Lassa fever, Marburg virus disease and Rift Valley fever. We designed non-infectious synthetic DNA oligonucleotide sequences incorporating primer binding sites suitable for five assays, and a T7 promotor site which was used to transcribe the sequence. Transcribed RNA was used as template in a dilution series, extracted and amplified with RT-PCR and RT-qPCR to demonstrate successful recovery and determine limits of detection in a range of laboratory settings. Our results show this approach is adaptable to any diagnostic assay requiring validation of nucleic acid extraction and/or amplification, particularly where sourcing reliable, safe material for positive controls is infeasible.

**Data Availability Statement:** The authors confirm that all data underlying the findings are fully available without restriction. All relevant data are within the paper and its Supporting Information files.

## Author summary

The majority of zoonoses originate in wildlife and tend to emerge from biodiverse regions in low to middle income countries, frequently among deprived populations of at-risk people with a lack of access to diagnostic capacity or surveillance. Diseases such as Crimean-Congo hemorrhagic fever, Rift Valley fever, Ebola virus disease, Marburg virus disease and Lassa fever are viral hemorrhagic fevers (VHFs) and rank among the most neglected and serious threats to global public health. This threat is partly due to the severity of disease caused by these pathogens, but also because their geographical distribution is close to human populations with often limited access to medical or diagnostic laboratory services. In our study we describe techniques for PCR based detection of five VHF viruses using a synthetic, multi-target non-infectious positive control. Our work has applications in assay design and optimization, particularly where access to source material is problematic or requires high level biosafety containment, as is the case with VHF viruses. This approach can help learners train in techniques used in nucleic acid extraction, amplification, and sequencing of VHF viruses and can be used for any targets, with potential for multiplexing from a single positive control.

**Funding:** This work was supported by funds from the World Organisation for Animal Health (Grant 3000034275; OIE Laboratory (or Collaborating Centre) Twinning Project: Enhancing capacity for early detection of viral haemorrhagic fevers in Liberia through epidemiological and laboratory training), Royal Society Te Apārangi Rutherford Discovery Fellowship (RDF-MAU1701) and the Percival Carmine Chair in Epidemiology and Public Health (all to DTSH). The funders had no role in study design, data collection and analysis, decision to publish, or preparation of the manuscript.

**Competing interests:** The authors have declared that no competing interests exist.

## Introduction

Medical advances including access to healthcare and sanitation have reduced infectious disease mortality and morbidity globally. However, infectious diseases remain a significant burden to many of the most deprived people in the world, and emerging infectious diseases (EIDs) are serious public health threats, as evidenced by the emergence of SARS-CoV-2 and COVID-19 pandemic in 2019 [1]. Most recorded human EIDs are zoonotic in origin, meaning they emerge from animals and cross the species barrier to infect humans. Factors that lead to EID emergence include socioeconomic factors, land use change, and urban population growth [2–5]. The majority of zoonoses (e.g. ~72% [3]) originate in wildlife and tend to emerge from biodiverse regions in low to middle income countries (LMICs) and frequently among deprived populations of at-risk people with a lack of access to diagnostic capacity or surveillance [6]. The recent global pandemic of monkeypox virus highlights the diagnostic issue further [7]; the virus that causes Mpox (monkeypox), an endemic zoonosis in West and Central Africa emerged locally, possibly in Nigeria, went undetected until it was detected in Europe [7].

Diseases such as Crimean-Congo hemorrhagic fever (CCHF), Rift Valley fever (RVF), Ebola virus disease (EVD), Marburg virus disease (MVD) and Lassa fever (LF) are viral hemorrhagic fevers (VHFs) and are among the most neglected and serious threats to global public health. This is in part due to the severity of disease caused by these pathogens, but also because their geographical distribution is close to human populations with often limited access to medical or diagnostic laboratory services. The large West African EVD outbreak beginning in 2013 was likely able to establish during the months after the first case because early cases were not detected in areas poorly served by diagnostic services [8]. There is therefore an urgent need to develop tools and in-country training methods for surveillance and diagnosis for both people and wildlife hosts for the VHFs [9]. Accordingly, we designed an assay for training purposes in Leon Quist Ledlum Central Veterinary Diagnostic Laboratory, Liberia as part of a WOAH (then OIE) twinning program to support personnel in surveillance methods for VHFs in animal hosts (https://rr-africa.woah.org/en/projects/ebo-sursy-en/).

Numerous PCR based diagnostic assays exist for the viral agents of these VHFs [10–15] providing valuable tools in disease surveillance and diagnosis. This relevance of PCR based approaches for clinical diagnosis, research and surveillance is despite the increasing use of rapid diagnostic tests, which typically have lower sensitivity and specificity compared to PCR, and because they are host-species agnostic, so able to be applied to samples from any species [16]. However, there are difficulties in training personnel, particularly for VHFs and pathogens that are both rare and highly pathogenic [17–19]. Developments in synthesizing nucleic acids, however, allow the synthesis of non-infectious control material for safe handling, including in low resource settings [20,21]. Here, we designed synthetic DNA oligonucleotide sequences incorporating primer binding sites suitable for five VHF (CCHF, RVF, EVD, MVD, LF) causing viruses including a T7 promotor site to transcribe the sequence to enable RNA synthesis and provide positive control material for all laboratory steps from nucleic acid extraction to detection. We tested the sensitivity of extraction using both RT-PCR (endpoint PCR) and RT-qPCR (quantitative PCR), including from samples simulated with spiked feces. In addition, since access to refrigeration and cold storage is often not possible in field settings, we assess the stability of reagents for amplification after storage at ambient temperature over a one-week time course.

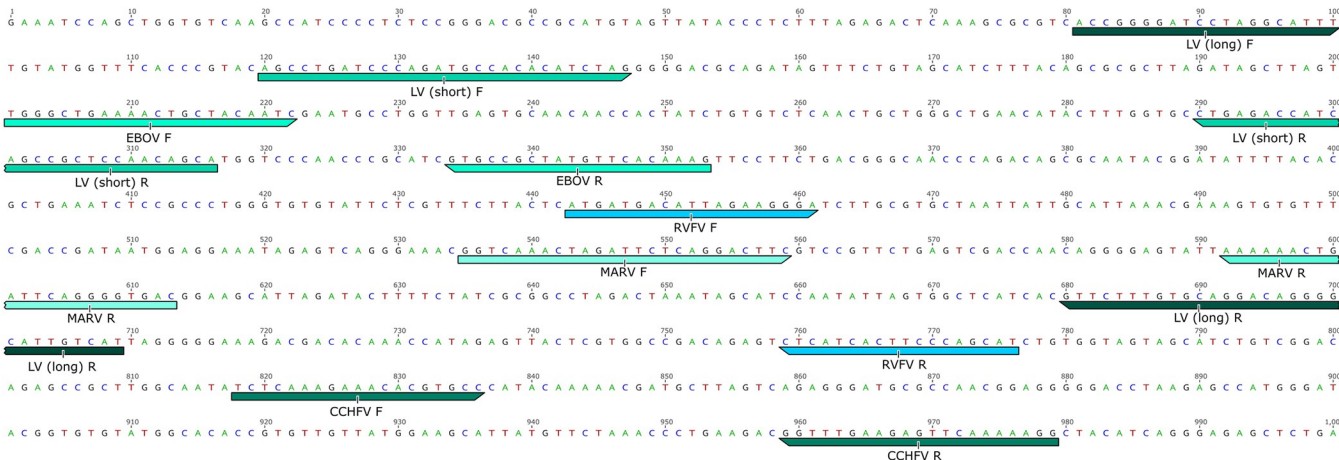

**Fig 1. Viral hemorrhagic fever synthetic insert with primer sequence locations.** See text and Table 1 for details.

## Methods

### VHF Control Vector

A 1,000 bp DNA fragment was designed by generating random nucleotide sequence and embedding previously published primer binding sites [10–15] within the sequence at positions designed to amplify the target amplicon size for each assay. The fragment contained primer binding positions for the amplification of five negative strand RNA viral pathogens, comprising three viruses from the order Bunyavirales: *Orthonairovirus haemorrhagiae* (Crimean-Congo hemorrhagic fever virus (CCHFV, family Nairoviridae)), *Mammarenavirus lassaense* (Lassa virus (LV), family Arenaviridae)), and *Phlebovirus riftense* (Rift Valley fever virus (RVFV), family Phenuiviridae)), and two viruses from the order Mononegavirales: *Orthoebolavirus zairense* (Ebola virus (EBOV), family Filoviridae)) and *Orthomarburgvirus marburgense* (Marburg virus (MARV), family Filoviridae)) (Fig 1, Table 1). PCR was performed with M13 primers to amplify and confirm the sequence of the vector insert.

The synthesised product was manufactured by GeneScript and cloned using pET-20b(+) vector, which includes a T7 promoter site (Fig 2). The resulting lyophilized plasmid DNA was reconstituted in 20 μL sterile water, but could be stored in the original, relatively stable dried form at room temperature for up to 3 months and at -20 for over a year. RNA copies of the fragment were generated from plasmid DNA using the MAXIscript T7 In Vitro Transcription kit (Ambion) following the manufacturer's instructions (including the optional DNAse digest).

**Table 1. Primers and probes for RT-PCR and RT-qPCR.**

| Virus* | Forward primer (5'-3') | Reverse primer (5'-3') | Size (bp) | Reference |
|---|---|---|---|---|
| CCHFV | TCTCAAAGAAACACGTGCC | CCTTTTTGAACTCTTCAAACC | 122 | Atkinson et al., 2012 (10) |
| EBOV | TGGGCTGAAAAYTGCTACAATC | CTTTGTGMACATASCGGCAC | 111 | Gibb et al., 2001 (12) |
| LV (long) | ACCGGGGATCCTAGGCATTT | ATGACMATGCCCCTKTCCTGCACAAAGAAC | 580 | Olschläger et al., 2010 (14) |
| LV (short) | AGCCTGATCCCAGATGCCACACATCTAG | TGCTGTTGGAGCGGCTGATGGTCTCAG | 197 | He et al., 2009 (15) |
| MARV | GGTCAAACTAGATTCTCAGGACTTC | GTCACCCCTGAATCAGTTTTTT | 80 | Huang et al 2012 (13) |
| RVFV | ATGATGACATT GAAGGGA | ATGCTGGGAAGTGATGAG | 298 | Garcia et al., 2001 (11) |
| β-globin internal control | AGAATCCAGATGCTCAAGGC | AGGTTCCTTTGTTCCCTAAGT | 72 | C. Bromhead; this study |

*CCHFV: Crimean-Congo hemorrhagic fever virus; EBOV: Ebola virus; LV: Lassa virus; MARV: Marburg virus, and: RVFV: Rift Valley fever virus.

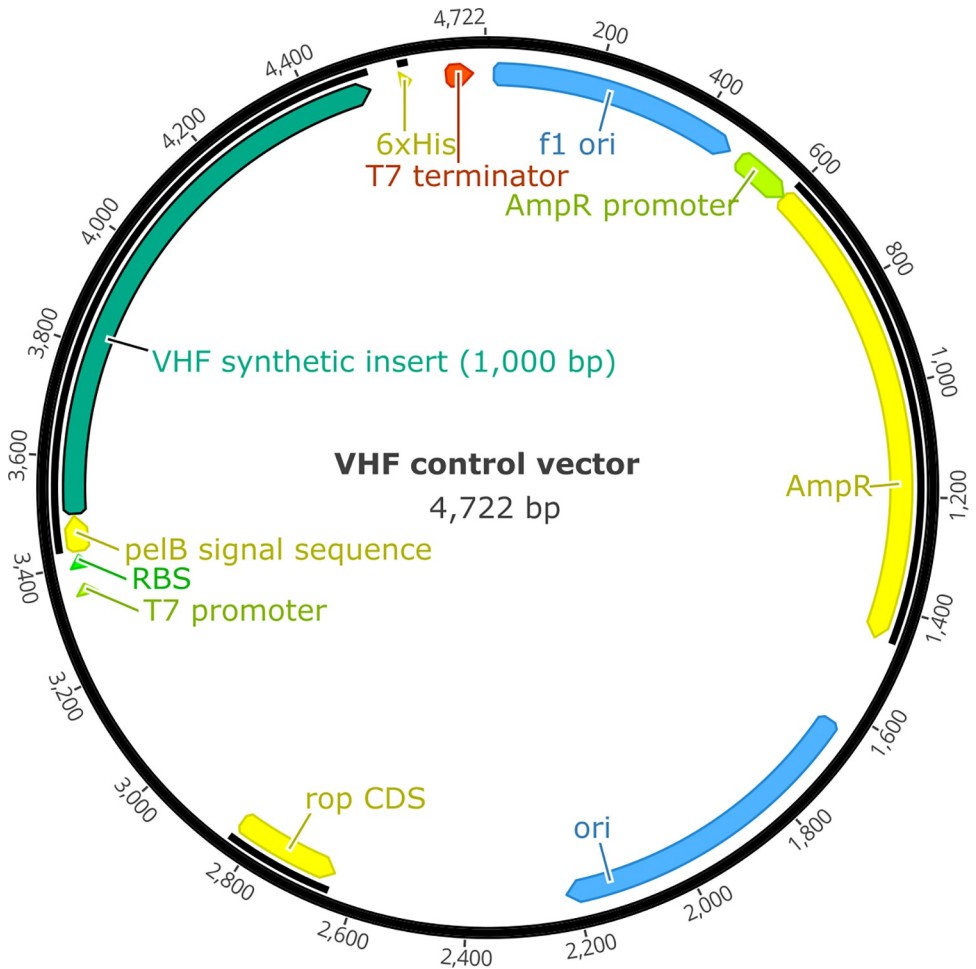

**Fig 2. pET-20b(+) vector including insert, which is the sequence in Fig 1.**

Synthesized RNA was then quantified using Qubit RNA HS (High Sensitivity) Assay Kit and stored at -80˚C until further analyses. We estimated the copy number based on the calculation which can be found at http://www.scienceprimer.com/copy-number-calculator-for-realtime-pcr.

### RNA extraction and 1-step RT-PCR

To simulate the biological sample processing steps, the synthetic VHF control RNA (550– $5.5 \times 10^{11}$ copies) was added to 1 mL sterile PBS or spiked in 900 μl PBS with 100 mg human fecal sample and then passed through a sterile 0.45 μm filter (Macherey-Nagel GmbH & Co. KG, Düren, Germany). We used feces as a biological test matrix because the complex composition of feces means that PCR inhibitors are more likely than in many other biological test matrices. We used 200 μL of filtrate as the input material for nucleic acid extraction using the High Pure Viral Nucleic Acid kit (Roche, New Zealand) according to the manufacturer's procedure. Extracted RNA was amplified by 1 step RT-PCR using each of the five VHF primer sets (Table 1) and assay-specific cycling protocols (Table 2). Each 20 μL reaction consisted of 0.25 μM each primer, 1 pg-1μg of template RNA (VHF Control Vector, GeneScript USA), 1 x PCR buffer and 0.5 μL SuperScript III RT/Platinum Taq High Fidelity Enzyme Mix

**Table 2. Cycling conditions for 1-step RT-PCR assays.**

| Virus* | Amplification** | | | Cycles |
|---|---|---|---|---|
| CCHFV | 95°C 10 s | 55°C 30 s | 68°C 30 s | 45 |
| EBOV | 94°C 15 s | 60°C 30 s | 68°C 30 s | 40 |
| LV (long) | 95°C 20 s | 55°C 20 s | 68°C 60 s | 45 |
| MARV | 94°C 15 s | 60°C 30 s | 68°C 30 s | 45 |
| RVFV | 95°C 15 s | 55°C 30 s | 68°C 30 s | 45 |

*CCHFV: Crimean-Congo hemorrhagic fever virus; EBOV: Ebola virus; LV: Lassa virus; MARV: Marburg virus, and: RVFV: Rift Valley fever virus.

**Note that all 1-step RT-PCR assays started with a cDNA synthesis step at 55°C for 30 min followed by an initial denaturation at 95°C for 2 min.

(Invitrogen). RT-PCR products were separated by agarose gel electrophoresis and visualised under UV light where bands of the expected size were identified and excised. DNA was eluted in 50 μL buffer (10 mM Tris, pH 8.0) for 12–24 hours at 4°C and then sent for bi-directional Sanger sequencing to the Massey Genome Service (Massey University, Palmerston North, New Zealand).

To test the sensitivity of the nucleic acid extraction and 1-step RT-PCR assays, we conducted a dilution series with the synthesized RNA. Dilutions for extraction were prepared in PCR grade water and ranged from 31.2 to $3.12 \times 10^{-8}$ ng (copy numbers $5.5 \times 10^{11}$ to 550) and followed the same biological sample processing steps as above.

## RT-qPCR assay and reagent stability experiments

All RT-qPCR assays used Ultraplex 1-Step ToughMix (Quantabio) and EvaGreen 20x (Biotium, both from DNature NZ Ltd) on a Roche LightCycler-96 instrument. We included two additional primer sets for RT-qPCR assays: a shorter Lassa virus amplicon [15] and a beta-globin internal cellularity control (β-globin) which we found could be duplexed with each VHF assay. Cycling conditions were optimised by testing annealing temperature gradients from 55°C to 65°C in duplicate for all seven assays under the following thermocycler conditions: 50°C for 10 mins, 95°C for 3 mins, 55 cycles of 95°C for 5 s, 55–65°C for 15 s, 72°C for 30 s, with fluorescence acquired during the extension step (excitation/emission EvaGreen = 488/ 530 nm). See Table 3 for optimal run conditions for each VHF. The average Cq value, the PCR cycle number at which sample reaction curve intersects the threshold line, of each duplicate was compared to find the optimal range of annealing temperatures, with the lowest Cq value chosen.

**Table 3. RT-qPCR VHF virus assay optimal experimental conditions and limits of detection.**

| Target* | Optimal annealing temperatures (°C) | TM of melting peaks (°C) | Limit of detection (copies/ μL) |
|---|---|---|---|
| EBOV | 55–61 | 82.5 | 50 |
| RVFV | 55–56 | 82.5 | 50 |
| CCHFV | 55–62 | 83 | 50 |
| MARV | 55–61 | 78 | 50 |
| LV-long | 55–65 | 77 | 500 |
| LV-short | 55–58 | 81 | 50 |
| β-globin | 55–60 | 81 | 10 |

* EBOV: Ebola virus, RVFV: Rift Valley fever virus, CCHFV: Crimean-Congo hemorrhagic fever virus, MARV: Marburg virus and LV: Lassa virus.

Primer concentrations were optimised by titration from 0.2 to 0.6 μM in 0.05 μM increments using a dilution series from 5000 copies/μL to 0.5 copies/μL of the synthetic control extracted as for test samples (High Pure Viral kit, Roche New Zealand). The dilution series was used to determine the limit of detection of each assay (see results, Table 3). Optimal reagent conditions for each VHF+ β-globin RT-qPCR assay, in a 10 μL reaction, consisted of 1 x Toughmix 1-step buffer, 1 x EvaGreen dye, 0.45 μM each primer (VHF + β-globin), synthetic RNA control (50–5000 copies/μL) and RNAse-free water. All the PCR tests above were run on sterile PBS extracted control material. To investigate the effect of potential inhibitors on PCR or extraction efficiency on detection thresholds we performed the CCHFV RT-qPCR assay on samples filtered through both PBS and spiked fecal samples. These tests were performed using the same RNA control template as used in the previous RT-qPCR assays after >1 year of storage at -80˚C.

To assess the performance of our assays under potential field conditions where refrigeration may not be available, we tested the performance of RT-qPCR reagents Ultraplex 1-Step Tough-Mix and EvaGreen 20x stored at room temperature (12–28˚C) for up to 1 week before use. Duplicate RT-qPCR tests for each VHF were performed at each of three timepoints (24 hours, 72 hours and 168 hours, 7 days) using the room temperature reagents. Synthesised RNA from the VHF Control Vector was used as the template at a concentration of 1,600–2,500 copies/μL.

## Results

### RT-PCR assay detection limits

RT-PCR performance varied among the assays (S1 Table, S1 Fig). The results of a dilution series of synthetic control from 5.5E+11 to 552 copies showed all five assays could detect to a limit of ~5.5 million copies/μL by gel electrophoresis, while the LV and MARV could reliably detect 5.5E+4 and gave visible weak bands at ~550 copies/μL. Sequencing of excised amplified products confirmed that recovered DNA matched target amplicons.

### RT-qPCR assay detection limits and time course experiment

The RT-qPCR assays underwent optimisation for annealing temperatures, primer concentrations and high-resolution melt (HRM) analysis. The results for these parameters as well as the limits of detection for each RT-qPCR assay is shown in Table 3 and Fig 3.

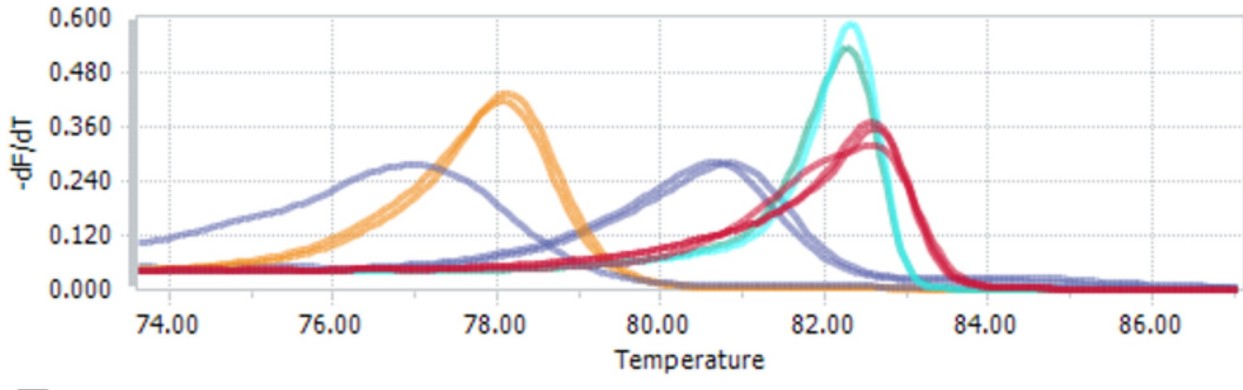

**Fig 3. High resolution melting peaks for Ebola virus and Crimean-Congo hemorrhagic fever (red), Rift Valley fever virus (turquoise), Marburg virus (yellow), short Lassa virus and β-globin (81˚C, purple), long Lassa virus (77˚C, purple).** Colours are grouped by peak melting temperature (˚C).

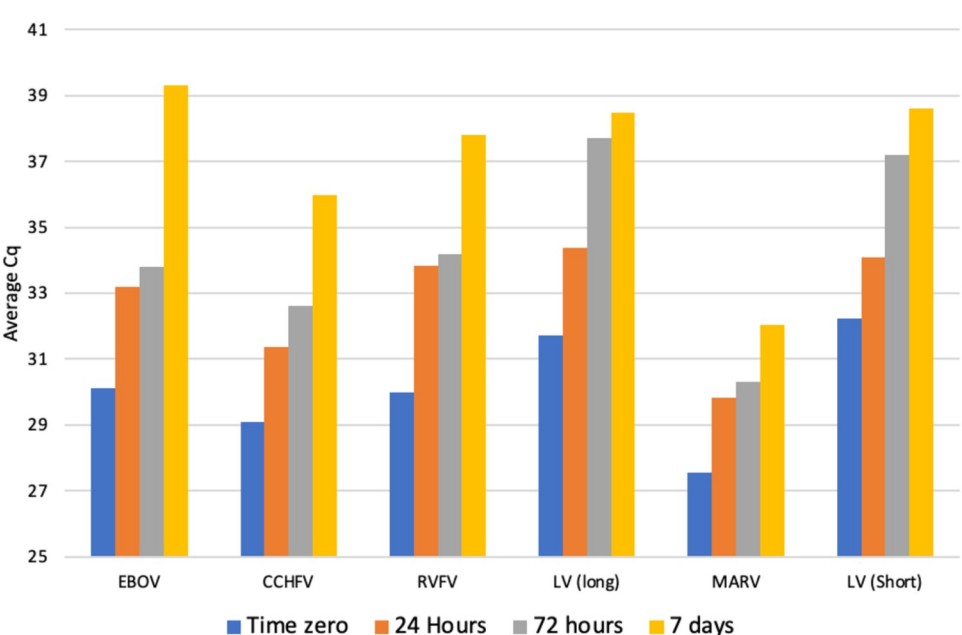

**Fig 4. Room temperature stability time series test of ToughMix and EvaGreen for VHF virus RT-qPCR assays.** Cq is the PCR cycle number at which sample reaction curve intersects the threshold line. EBOV: Ebola virus, CCHFV: Crimean-Congo hemorrhagic fever virus, RVFV: Rift Valley fever virus, LV: Lassa virus and MARV: Marburg virus.

In our limit of detection experiments, all RT-qPCR assays could reliably detect 50 copies per μL, except for the long LV assay which could only detect down to 500 copies/μL consistently. The decreased efficiency of this assay indicated it was not suitable for qPCR conversion. No change in primer concentration (0.45 mM) was required for combining any of the VHF assays with the β-globin internal control and a standard annealing temperature of 56˚C was found to suit all assays. Sterile PBS extractions were found to amplify slightly earlier (2–2.6 cycles) than fecal spiked samples at high template concentrations (100,000 copies) one year after storage, representing nearly a 10-fold difference in the limit of detection, but showed broadly analogous assay sensitivity, with an average Ct of 36.4 (PBS) and 36.7 (feces) for 10,000 copies (S2 Table). Comparison of these results with the original limit of detection results (Cq 29.1 using 1000 copies) showed a reduction in sensitivity (Cq 29.2 at 1.0E+6 copies) demonstrating that long term storage or RNA template and freeze thaw cycles will affect assay sensitivity.

We tested the performance of our VHF RT-qPCR assays using reagents stored at room temperature (12–28˚C), in the dark, for up to 1 week. Duplicate RT-qPCR tests for each VHF were performed at each of three timepoints (24 hours, 72 hours and 7 days) using the synthetic RNA control at a concentration of 1,600–2,500 copies/μL. The average Cq values at 24 hours, 72 hours and 1 week all rose compared to time zero (using ideal frozen reagents). The VHF virus RT-qPCR's biggest change was between 72 hours and 1 week for each VHF virus assay (Fig 4). The average Cq values were highest for the LV assays, particularly the long LV, indicating it was the most sensitive to reduced efficiency reagents. By contrast, the MARV assay maintained a low Cq across all three time points. After 1 week, the assay sensitivity for detecting 1000 copies/μL is significantly compromised. Therefore 3 days (72 hours) at room temperature is the maximum recommended storage time for Ultraplex 1-Step ToughMix (Quantabio) and EvaGreen 20x (Biotium) for this testing purpose (See S3 Table for the raw Cq data).

## Discussion

We have developed a single synthetic DNA oligonucleotide with a promotor site to generate positive controls for six assays for the most commonly occurring viral causes of VHF's using both RT-PCR and RT-qPCR, incorporating an internal control for the latter, with the possibility of multiplexing and using reagents stored at room temperature for up to 72 hours.

Our results demonstrate the applicability of synthetically designed nucleic acids for use as molecular diagnostic assay controls in both RT-PCR and RT-qPCR methods. This work has applications for assay design and optimization, particularly where access to source material is difficult to obtain or requires high level biosafety containment, as is the case with VHF viruses. The control material is clearly distinguishable from actual viral samples using sequence analysis, allowing easy detection of cross contamination. This approach can help learners to train in techniques used in nucleic acid extraction, amplification using either analogue or quantitative PCR, and sequencing of VHF viruses.

As well as providing ongoing validation for the VHF viral assays we present, this approach can be used for any targets, so enabling learners from any countries, including LMICs most burdened by infectious diseases, to design custom assay controls specific to local disease burdens, with potential for multiplexing from a single positive control. Developing new targets requires knowledge and researchers may need training, access to software and the ability to import reagents, but is a cheap and safe way to create stable control material. We had no problems detecting DNA or RNA from the synthetic vector insert. However, RNA secondary structures may impede amplification in some cases, so some care may be needed in the synthetic insert design for other targets.

Our assay was designed for use as a training tool and help test reagent viability, from extraction steps to PCR and sequencing, but it also performed well in the presence of potential inhibitors in human feces, which are commonly found in real samples. Other sample matrices such as blood or urine were not tested as they are considered less challenging for PCR inhibitors, but further testing may be advisable depending on the applications of future studies. Furthermore, to better test performance and detectability in other biological sample matrices further studies could use control material encapsulated with a pseudovirus or proteinaceous shell [22], as well as measure how matrices impact RNA purity. Future studies can also explore the plasmid vector and reagent stability better and under different conditions [23,24]. Moreover, further studies using synthetically designed control material may also consider the incorporation of probe-based RT-qPCR. This could be achieved by adding a probe sequence to the appropriate site within the amplicon and running the PCR with probes rather than fluorescent dye, though it is more costly. Additionally, if the amplicon regions were designed as linear rather than overlapping amplicons, the synthetic assay design could directly match the target sequence allowing high resolution melt curve analyses. However, any of these detection strategies requires validation by sequencing, which itself introduces challenges in low resource settings and additional training of researchers.

Recent technological advances have greatly improved virus detection and diagnosis without the need for multi-step RNA purification [25]. New generation RT-qPCR reagents are more robust to temperature storage above -20°C. These tests provide rapid, inexpensive, and robust diagnoses where laboratory infrastructure is not available, and may replace current technologies (i.e., RT-PCR assays) in some settings. However, there is an urgent need to develop in-country tools for surveillance and diagnosis for both people and wildlife hosts for the VHFs [9]. Since our workflow requires only standard equipment, currently present in most molecular biology laboratories, it has applications for the design and validation of tests used in surveillance of VHFs in countries where such work is most needed and for the time being RT-PCR

and RT-qPCR are likely to remain the most common method for many diagnostic laboratories. Therefore, safe approaches to generate control material and to train staff are needed and our work provides further evidence for the applicability of synthetic nucleic acids for use as assay controls. While this assay was originally designed with a specific training application for detecting VHFs in Liberia, similar approaches could be used for training and surveillance in similar circumstances to detect outbreaks of infection in animal populations where laboratories may not be as well-resourced as in human clinical settings, or more regional laboratories in lower-resource settings in LMICs.

## Supporting information

**S1 Fig. RT-PCR gel results from dilution series experiment on five VHF virus assays.** Lanes are labelled as follows: A: 1kb ladder (expanded on right of figure), B: $5.5 \times 10^{12}$ copies, C: $5.5 \times 10^{11}$ copies, D: $5.5 \times 10^{9}$ copies, E: $5.5 \times 10^{7}$ copies, F: $5.5 \times 10^{5}$ copies, G: $5.5 \times 10^{3}$ copies, H: negative control. CCHFV: Crimean-Congo hemorrhagic fever virus, EBOV: Ebola virus, LV: Lassa virus, MARV: Marburg virus and RVFV: Rift Valley fever virus.
(DOCX)

**S1 Table. Gel electrophoresis detection results for a dilution series experiment on five viral hemorrhagic fever virus assays.**
(DOCX)

**S2 Table. Comparison of sample matrices (PBS v fecal) on RT-qPCR efficiency using the** *Crimean-Congo hemorrhagic fever virus* **(CCHFV) assay.**
(DOCX)

**S3 Table. Raw Cq values from time course experiments.**
(DOCX)

## Acknowledgments

We wish to acknowledge the valuable contributions of WOAH staff, particularly Sophie Muset and Mariana Marrana, and the Central Veterinary Laboratory team in Liberia.

## Author Contributions

**Conceptualization:** David TS Hayman.

**Formal analysis:** Matthew A. Knox, Collette Bromhead.

**Funding acquisition:** David TS Hayman.

**Investigation:** Matthew A. Knox, Collette Bromhead.

**Methodology:** Matthew A. Knox, Collette Bromhead.

**Project administration:** David TS Hayman.

**Validation:** Collette Bromhead.

**Visualization:** Matthew A. Knox.

**Writing – original draft:** Matthew A. Knox.

**Writing – review & editing:** Collette Bromhead, David TS Hayman.

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
