## [Decision Letter · Decision Letter 0]

25 Sep 2023

Dear Dr. Matthew A Knox,

Thank you very much for submitting your manuscript "Development of a non-infectious control for viral hemorrhagic fever PCR assays" for consideration at PLOS Neglected Tropical Diseases. As with all papers reviewed by the journal, your manuscript was reviewed by members of the editorial board and by several independent reviewers. In light of the reviews (below this email), we would like to invite the resubmission of a significantly-revised version that takes into account the reviewers' comments. 

We cannot make any decision about publication until we have seen the revised manuscript and your response to the reviewers' comments. Your revised manuscript is also likely to be sent to reviewers for further evaluation.

Sincerely,

Aparna Krishnavajhala, Ph.D.

Academic Editor

Elvina Viennet

Section Editor

Reviewer's Responses to Questions

**Key Review Criteria Required for Acceptance?**

**Methods**

-Are the objectives of the study clearly articulated with a clear testable hypothesis stated?

-Is the study design appropriate to address the stated objectives?

-Is the population clearly described and appropriate for the hypothesis being tested?

-Is the sample size sufficient to ensure adequate power to address the hypothesis being tested?

-Were correct statistical analysis used to support conclusions?

-Are there concerns about ethical or regulatory requirements being met?

Reviewer #1: In their manuscript, Knox and colleagues present the development of synthetic DNA oligonucleotide sequences encoding primer binding sites of different viral haemorrhagic fever (VHF) viruses and their application in conventional and real-time RT-PCRs with the aim of implementing this strategy under field conditions. Generally, the development of in-vitro transcribed RNA as controls for various PCR assays is not entirely new. However, the general idea to establish protocols that support capacity building in areas where access to diagnostic laboratory services has been limited, is crucial to strengthen the laboratory infrastructure of the respective countries. Moreover, this study underlines the importance of epidemiological surveillance studies in these areas and the need for reliable diagnostics and well-trained personnel.

Although the general performance of the in-vitro transcribed RNA seems promising, there are some shortcomings in the overall methodology that should be addressed. The description of some important points was insufficient or completely missing. 

Some major points:

p. 8, lines 89 ff.: Taxonomy. It seems that there is no unified way of using the virus taxonomy. Some viruses are written in italics, others not. For some, the species name was used (CCHF orthonairovirus and RVF phlebovirus). Here, one could go for the virus name like in Ebola, Lassa and Marburg virus. Only when referring to the species/genus/family, the name is written in italics. “[…] order Bunyavirales (italics!): Crimean-Congo haemorrhagic fever virus (non-italics!) (CCHFV; Crimean-Congo haemorrhagic fever orthonairovirus (italics!), family Nairoviridae (italics!), […]” and so on. It starts in the abstract and is continued in the Methods section.

p. 8, line 85: how randomly generated is this sequence between the primer binding sites? Is there anything to consider when generating it (e.g. RNA secondary structure, anything that could hamper RNA transcription)? How are the melting peaks affected by sequence design? 

p. 8, line 87: the fragment contains primer binding positions. What about using probes in addition? Conventional PCR and melting curve analyses-based qPCR requires sequencing to avoid misinterpretation/to exclude contamination. Certainly, this can also happen in probe-based qPCRs. However, under field settings, sequencing may be a challenge. Also, melting curve analyses require intensive training of personnel. At the very least, please discuss advantages and disadvantages of all these methods.

p.10, line 109: does manufacturer’s instruction include DNAse digest? It would be helpful to mention it and discuss its importance when discussing the results. When setting up all the assays, have PCRs been performed without reverse transcription? Was there an amplification? How did the authors determine purity of RNA?

Reviewer #2: Most clinical laboratories in VHF-prone countries in sub-Saharan Africa use real-time PCR to diagnose EBOV and LASV infection. What is the rationale for using analogue RT-PCR in this paper?

Reviewer #3: In this manuscript Knox et al describe their development of a non-infectious control for viral hemorrhagic fever PCR assays. The article is well-written and results are clearly presented. Overall the idea and preliminary results with the control(s) appear promising. However, the authors overstate or misunderstand the application of the controls in their current state of development. Are the controls for use in the field or a molecular biology lab? If the former, then how do you operate a quality-equivalent thermal cycler in an area where "access to refrigeration and cold storage is often not possible"?. If the latter, then why is so much emphasis put in the manuscript about low-resource settings? The control as currently presented/validated is more appropriate for developed world diagnostic laboratories where indeed lab staff from low-resource countries could be trained. 

Scientifically, the biggest concern I see with the control validation is the fact that experiments are always being performed in sterile PBS, sterile water, PCR grade water, and RNase-free water. Minimally the authors should present results with fresh blood samples spiked with the controls. Ideally the control would be encapsulated with a pseudovirus or proteinaceous shell.

Another problem with the validation is the idea that 28C is an adequate challenge for RNA for these VHFs, many of which occur in the middle east, Sahara, West Africa, and equatorial Africa where average daily temperatures are around 27C.

Some minor comments:

Line 41: is "but" or "and" more appropriate?

Line 51: change animal to animals.

Lines 53-56: Need reference for this statement.

Line 58: This sort of "must" language should be removed from scientific publication.

**Results**

-Does the analysis presented match the analysis plan?

-Are the results clearly and completely presented?

-Are the figures (Tables, Images) of sufficient quality for clarity?

Reviewer #1: Some major points:

p. 13, line 169 ff.: the detection limit is around 5 million copies per µl for some of the conventional PCRs, that seems a lot. What about the dilutions of 1:10^4, 1:10^6 and 1:10^8? Have those been performed and tested? It would give more accuracy to the detection limit and maybe even improve performance of the PCRs. Especially when comparing it to the qPCR data where 50 copies were sufficient. This is round about 5 log. Have the authors tested the effect different primer concentrations may have? That would be important to know at this point. 

For the supplementary Figure 1, it would help to have the DNA ladder size indicated, maybe at least for some of the bands.

p. 13, line 180: Table 3: the optimal annealing temperature has a broad range for almost all the assays. How was this determined? Please describe. What were the quality criteria to consider the temperature optimal? P. 14, l. 192, please mention the criteria to decide for 56°C annealing temperature.

p. 14, line 184: Figure 3, if the synthetic construct does not include the authentic sequence amplified (but only a random sequence as mentioned above), how does that affect the melting peaks when comparing it with that of an (authentic) field sequence? Please include at least a statement in your discussion.

p. 15, line 208: please add abbreviations to the figure legend. What does Cq mean, what is displayed at all? The figure legend is a little short in information.

p.16, line 229 ff., please also discuss the capacity building needed to train personnel in designing a synthetic construct (and possibly the difficulty to obtain synthesized sequences in some low to middle income countries).

Reviewer #2: RT-PCR and RT-qPCR

• The authors have not demonstrated the relevance of these RT-PCR and RT-qPCR conditions by showing that they can successfully amplify nucleic acids from any of the five different VHF pathogens in real or spiked samples.

Stability of PCR reagents

• This section is poorly designed. 

• Why are the authors testing the stability ToughMix and Eva Green reagents, when the focus of the paper is on the synthetic positive control material? One of the main reasons that diagnostic PCR reactions fail in the field is due to degradation of the positive control nucleic acids. What the authors should have tested is the stability of their synthetic control material, at different temperature, through different number of freeze-thaw cycles. 

• The ambient temperature used (12-28°C) is inadequate. The ambient temperature in West Africa can reach 38°C in the dry season. 

• There is no control. Would the Cq also increase after 7 days if the reagents were stored at 2-8°C or -20°C?

Reviewer #3: (No Response)

**Conclusions**

-Are the conclusions supported by the data presented?

-Are the limitations of analysis clearly described?

-Do the authors discuss how these data can be helpful to advance our understanding of the topic under study?

-Is public health relevance addressed?

Reviewer #1: see above suggestions concerning the improvement of the discussion.

Reviewer #2: The study is poorly designed and lacks key controls.

The results do not fully demonstrate relevance or application.

Reviewer #3: (No Response)

**Editorial and Data Presentation Modifications?**

Reviewer #1: (No Response)

Reviewer #2: Line 92

The correct abbreviation for Ebola virus is EBOV.

Line 93

The correct abbreviation for Marburg virus is MARV.

Reviewer #3: (No Response)

**Summary and General Comments**

Reviewer #1: try Taqman-based/probe-based PCR in comparison with conventional and melting curve-based PCR.

complete dilution series (10^4/6/8 were missing)

show experiments to determine annealing temperature

Reviewer #2: The development of non-infectious control materials for VHF diagnosis is welcome. But the result presented in this paper is not complete. The experimental design for reagent stability is flawed. The use of the invented reagent has not been fully validated.

Reviewer #3: (No Response)

PLOS authors have the option to publish the peer review history of their article (what does this mean?). If published, this will include your full peer review and any attached files.

Reviewer #1: No

Reviewer #2: No

Reviewer #3: No
---

## [Decision Letter · Decision Letter 1]

4 Mar 2024

Dear Matthew Knox,

Thank you very much for submitting your manuscript "Development of a non-infectious control for viral hemorrhagic fever PCR assays" for consideration at PLOS Neglected Tropical Diseases. As with all papers reviewed by the journal, your manuscript was reviewed by members of the editorial board and by several independent reviewers. In light of the reviews (below this email), we would like to invite the resubmission of a significantly-revised version that takes into account the reviewers' comments. 

We cannot make any decision about publication until we have seen the revised manuscript and your response to the reviewers' comments. Your revised manuscript is also likely to be sent to reviewers for further evaluation.

Sincerely,

Aparna Krishnavajhala, Ph.D.

Academic Editor

Elvina Viennet

Section Editor

Reviewer's Responses to Questions

**Key Review Criteria Required for Acceptance?**

**Methods**

-Are the objectives of the study clearly articulated with a clear testable hypothesis stated?

-Is the study design appropriate to address the stated objectives?

-Is the population clearly described and appropriate for the hypothesis being tested?

-Is the sample size sufficient to ensure adequate power to address the hypothesis being tested?

-Were correct statistical analysis used to support conclusions?

-Are there concerns about ethical or regulatory requirements being met?

Reviewer #4: In the manuscript, the authors present the development of synthetic DNA, infectious control for the viral hemorrhagic fevers PCR assays, this is a good subject, as for a lot of viral infections it is hard to extract the positive controls, particularly where the access to the source material is problematic or requires high level biosafety containment.

there are some grammatical errors in line31 and needs to be changed to "are". the same in line62.

Reviewer #5: The study only evaluated fecal samples, but should include other sample types unless other parts of the manuscript are updated to clearly reflect this is only one sample type. 

The authors mention that the lyophilized plasmid DNA could be stored in the relatively stable dried form, but include no experiments that indicate for how long and at what temps it can be stored and still successfully used in this assay. This is important given they highlight that this is a good tool for use in LMICs.

**Results**

-Does the analysis presented match the analysis plan?

-Are the results clearly and completely presented?

-Are the figures (Tables, Images) of sufficient quality for clarity?

Reviewer #4: The authors have not mentioned the importance of Rt PCR and RT qPCR. 

the figures are of sufficient quality.

Reviewer #5: Figures images are low quality and difficult to read but content is valuable.

**Conclusions**

-Are the conclusions supported by the data presented?

-Are the limitations of analysis clearly described?

-Do the authors discuss how these data can be helpful to advance our understanding of the topic under study?

-Is public health relevance addressed?

Reviewer #4: (No Response)

Reviewer #5: Limitations of testing only one sample type should be further discussed. Public health relevance is well addressed. Clarity edits are needed.

**Editorial and Data Presentation Modifications?**

Reviewer #4: (No Response)

Reviewer #5: General clarity edits are necessary. The text can be modified to be clearer and more direct.

**Summary and General Comments**

Reviewer #4: the paper has a good relevance but needs some minor clarifications and modifications.

Reviewer #5: The presented technology is valuable and useful, but the analyses of its use lacks important controls (alternate sample types, additional temperature storage testing) that need to be addressed before this can be presented as widely applicable controls.

PLOS authors have the option to publish the peer review history of their article (what does this mean?). If published, this will include your full peer review and any attached files.

Reviewer #4: No

Reviewer #5: No
---

## [Editor Report · Decision Letter 2]

13 Apr 2024

Dear Dr Knox,

We are pleased to inform you that your manuscript 'Development of a non-infectious control for viral hemorrhagic fever PCR assays' has been provisionally accepted for publication in PLOS Neglected Tropical Diseases.

Best regards,

Elvina Viennet, PhD

Section Editor

Elvina Viennet

Section Editor

---

## [Editor Report · Acceptance letter]

17 Apr 2024

Dear Dr Knox,

We are delighted to inform you that your manuscript, "Development of a non-infectious control for viral hemorrhagic fever PCR assays," has been formally accepted for publication in PLOS Neglected Tropical Diseases.

Best regards,

Shaden Kamhawi

co-Editor-in-Chief

Paul Brindley

co-Editor-in-Chief
